# Plasmatic Klotho and FGF23 Levels as Biomarkers of CKD-Associated Cardiac Disease in Type 2 Diabetic Patients

**DOI:** 10.3390/ijms20071536

**Published:** 2019-03-27

**Authors:** Ana Paula Silva, Filipa Mendes, Eduarda Carias, Rui Baptista Gonçalves, André Fragoso, Carolina Dias, Nelson Tavares, Hugo Mendonça Café, Nélio Santos, Fátima Rato, Pedro Leão Neves, Edgar Almeida

**Affiliations:** 1Nephrology Department, Centro Hospitalar Universitário do Algarve, 800-836 Faro, Portugal; filipabritomendes@gmail.com (F.M.); eduardaccarias@gmail.com (E.C.); matinhosfragoso@gmail.com (A.F.); pleaon@hotmail.com (P.L.N.); 2Departamento de Ciências Biomédicas e Medicina, Universidade do Algarve, 8005-139 Faro, Portugal; rui.baptistagoncalves@gmail.com (R.B.G.); diascarolinaj@gmail.com (C.D.); 3Cardiology Department, Centro Hospitalar Universitário do Algarve, 8000-386 Faro, Portugal; nelson.tavares63@gmail.com (N.T.); hu.cafe@gmail.com (H.M.C.); 4Clinic Pathology Department, Centro Hospitalar Universitário do Algarve, 8000-836, Faro, Portugal; neliofilipe.santos@gmail.com (N.S.); fatima.rato@gmail.com (F.R.); 5Faculdadade de Medicina da Universidade de Lisboa, 1600-190 Lisboa, Portugal; edealmeida@mail.telepac.pt

**Keywords:** klotho, FGF-23, CKD, CVD, LVMI

## Abstract

Background: Research over the past decade has focused on the role of Klotho as a cardio protective agent that prevents the effects of aging on the heart and reduces the burden of cardiovascular disease CVD. The role of the interaction between fibroblast growth factor 23-(FGF-23)/Klotho in Klotho-mediated actions is still under debate. The main objective was to ascertain the potential use of plasmatic Klotho and FGF23 as markers for CKD-associated cardiac disease and mortality. Methods: This was a prospective analysis conducted in an outpatient diabetic nephropathy clinic, enrolling 107 diabetic patients with stage 2–3 CKD. Patients were divided into three groups according to their left ventricular mass index and relative wall thickness. Results: Multinomial regression analysis demonstrated that low Klotho and higher FGF-23 levels were linked to a greater risk of concentric hypertrophy. In the generalized linear model (GLM), Klotho, FGF-23 and cardiac geometry groups were statistically significant as independent variables of cardiovascular hospitalization (*p* = 0.007). According to the Cox regression model, fatal cardiovascular events were associated with the following cardiac geometric classifications; eccentric hypertrophy (*p* = 0.050); concentric hypertrophy (*p* = 0.041), and serum phosphate ≥ 3.6 mg/dL (*p* = 0.025), FGF-23 ≥ 168 (*p* = 0.0149), α-klotho < 313 (*p* = 0.044). Conclusions: In our population, Klotho and FGF23 are associated with cardiovascular risk in the early stages of CKD.

## 1. Introduction

Aging is the result of a complex interplay between genetics, environment and lifestyle. It encompasses intricate body physiologic alterations that hamper organs’ function and compromises their long-term viability. As major regulators of many vital physiologic processes, the kidneys are often the first organs to express the effects of aging. Reduced glomerular filtration rate (GFR) and/or diminished production of several hormones and communication mediators usually appear first and are the triggers of major consequent alterations that frequently result in chronic kidney disease (CKD), anemia, osteoporosis and left ventricular hypertrophy (LVH) [1,2,3,4]. 

CKD is known to be related to cardiac overload, which reconfigures the architecture and physiology of the myocardium, inducing hypertrophy and fibrosis [4,5]. These alterations appear in the early stages of the disease and are aggravated with worsening kidney disease, compromising cardiac function and inducing the development of cardiovascular disease (CVD) [5]. Nowadays, kidney-associated CVD is a major public health concern in western societies, representing the ninth leading cause of death in the USA [6]. Markers of prognosis are needed, as are effective therapeutic strategies aimed at blocking these degrading cascades.

Research over the past decade has focused on the role of Klotho as a cardio protective agent that prevents the effects of aging on the heart and reduces the burden of CVD [7,8]. Whether the interaction of FGF23 and Klotho is essential for Klotho-mediated actions is still under debate, and clarification is needed as reports in the literature are inconsistent [9,10,11]. In line with the recent work of Drew and collaborators [12], the present work used a pool of 107 CKD patients to ascertain the potential use of plasmatic Klotho, FGF23 or both, as markers for CKD-associated cardiac disease and to discuss their eventual use as therapeutic targets in CKD. A secondary objective was to evaluate the role of mineral metabolism markers of the left ventricle, such as FGF-23 and α-Klotho in geometric form.

## 2. Results

This study enrolled one hundred and seven (107) consenting patients with stage 2–3 CKD, who were cleared of all exclusion criteria. The mean age was 57.2 ± 7.1 years (range: 41–68) and the mean LVMI level was 99.31 ± 23.45 g/m^2^ (range: 67–189). 

The demographic and clinical parameters of the study subjects are presented in Table 1.

In terms of allocation of patients into groups according to their LVH and RWT values, 41 (38.3%) had normal geometry, 38 (35.5%) had eccentric hypertrophy and 28 (26.2%) had concentric hypertrophy.

All the assessed variables, except age and BMI, showed significant differences between the three groups (Table 2).

A simple linear regression analysis (Table 3) revealed that LVMI levels were inversely proportional to α-Klotho (*r* = −0.440, *p* = 0.0001), and directly proportional to Pi (*r* = 0.672, *p* = 0.005), and to FGF-23 (*r* = 0.622, *p* = 0.0001).

Multinomial regression analysis demonstrated that diminished eGFR and elevated Pi levels were associated with a greater risk of eccentric hypertrophy but not with concentric hypertrophy. In fact, hyperphosphatemia seemed to increase the risk of eccentric hypertrophy threefold. Contrarily, low α-Klotho and higher FGF-23 levels were linked with a greater risk of concentric hypertrophy, with higher Klotho levels associated with a 26% lower risk (Table 4).

Figure 1 shows Kaplan–Meier survival curves in patients with normal left ventricular geometry, eccentric hypertrophy and concentric hypertrophy. A survival table is also given with the number of patients at risk. Censored subjects are indicated as tick marks. Comparison between the groups using the log-rank test revealed the highest risk group is with concentric hypertrophy (log rank = 9.422; *p* = 0.009). In our study, the cause of mortality was coronary and cerebrovascular disease in eccentric and concentric hypertrophy.

In the generalized linear model (GLM), Klotho, FGF-23 and cardiac geometry groups were statistically significant as independent variables of cardiovascular hospitalization (*p* = 0.007). The results showed that the α-Klotho group (<313), compared to the group at ≥313, had higher odds of hospitalization ORa = 1.491, *p* = 0.014. In addition, the FGF-23 group (≥168), compared to the group at <168, had higher odds of hospitalization, ORa = 1.689, *p* = 0.004. The concentric hypertrophy group compared to the no hypertrophy group had higher odds of hospitalization, ORa = 4.889, *p* = 0.023. The optimized model as a function of α-Klotho groups, eGFR, age, gender, phosphorus, PTH, IL6, vitamin D, FGF-23 and cardiac geometry groups allowed for a final and statistically significant model (*p* < 0.001), which included Klotho groups, FGF-23 and cardiac geometry variables. The area under the receiver operating characteristic curve (ROC curve) for this optimized model was 0.980 (*p* < 0.001), revealing an elevated discriminating capacity of the model. The obtained estimates showed that the group with low levels of α-Klotho (ORa = 1.320, *p* = 0.024), higher levels of FGF-23 (ORa = 1.105, *p* = 0.012) and the concentric hypertrophy group (ORa = 2.284, *p* = 0.041) had greater odds of cardiovascular hospitalization (Table 5).

According to Cox regression models (Table 6), fatal cardiovascular events were associated with the following cardiac geometric classifications; eccentric hypertrophy (*p* = 0.050); concentric hypertrophy (*p* = 0.041), and Pi ≥ 3.6 (*p* = 0.025), FGF-23 ≥ 168 (*p* = 0.0149), α-klotho < 313 (*p* = 0.044).

ROC curve analysis demonstrated that α-Klotho < 313 and FGF-23 ≥ 168 are the best cut-off values associated with cardiovascular mortality (AUC = 0.793; 95% CI 0.695–0.892), (AUC = 0.758; 95% CI 0.638–0.879), respectively.

## 3. Discussion

The current study shows that both Klotho and FGF23 are independently altered and associated with concentric hypertrophy in CKD patients, thus being suitable as indicators of cardiac disease in this population. Particularly, it provides evidence that CKD-associated cardiac hypertrophy correlates with reduced eGFR and soluble Klotho and Vitamin D levels, and is indicative of a generalized pro-inflammatory status, as evidenced by increased levels of IL-6, ACR, PTH, P and FGF23. Furthermore, it shows that concentric hypertrophy in particular is associated with a more severe phenotype with a shorter survival rate.

CKD is a very well-established inducer of cardiac hypertrophy, as it increases the heart burden in an attempt to accommodate kidney failure (recently reviewed by Lullo et al.) [13]. Aiming to correlate the alteration of the LV geometry with sudden cardiac death risk, Lang and collaborators proposed a 4-pattern classification of LV geometry based on both LVMI and relative wall thickness (normal geometry, concentric remodeling, eccentric hypertrophy and concentric hypertrophy) [14]. According to this classification, concentric hypertrophy confers the highest risk, followed by eccentric hypertrophy and concentric remodeling [15]. Therefore, our study is likely to be the first to corroborate the 4-pattern cardiac hypertrophy classification in a population of CKD patients.

As recently argued by Neyra and Hu, there is an urgent need to identify sensitive and/or more specific diagnostic and prognostic biomarkers for CKD [16]. In fact, while very sensitive as an indicator of cardiac hypertrophy, the 4-pattern classification is very unspecific and must be combined with other parameters to provide clues on the etiology of the cardiac remodeling. In this line, our results substantiate others in the literature showing that Klotho and vitamin D are consistently reduced in CKD patients, while Pi, FGF-23 and PTH are consistently elevated [17,18,19,20,21,22]. Moreover, and unlike what was observed by Tanaka and collaborators, these associations were significant in all CKD patients, particularly between Klotho and cardiac hypertrophy [11]. Altogether, these data support the interpretation of the LV morphological results in light of the abovementioned physiological parameters as both diagnostic biomarkers of CKD and useful follow up parameters in these patients. They have been proved to not only have diagnostic value but also sensitive prognostic potential. For instance, mortality data previously collected revealed significant differences among the three cardiac geometry groups. Patients with concentric hypertrophy tended to have worse survival rates than patients without left ventricular hypertrophy. We demonstrated that, after adjustment for cofounders, CKD patients with low serum α-Klotho levels (≤313 pg/mL) and high serum FGF-23 levels (≥168 pg/mL) are at high risk of adverse cardiovascular outcomes.

Furthermore, in the generalized linear model, concentric hypertrophy was found to independently predict patient survival, with increased risk of hospitalization [23,24,25].

Regarding the interaction between α-Klotho and FGF-23, our results seem to corroborate those of Grabner et al., suggesting independent roles for each protein. These authors demonstrated that FGF-23-mediated activation of FGFR4 in cardiomyocytes stimulates phospholipase Cc/calcineurin-NFAT signaling, independently triggering LV hypertrophy [26]. Moreover, and in agreement with this observation, both Faul et al. and Shibata et al. reported a statistically significant association between serum FGF-23 levels, LV hypertrophy and cardiac systolic function [17,27], mediated by an increase in α-actin in cardiac muscle cells, with an increase in expression of LVH markers, such as fetal heavy chain β-myosin, and a drop in adult heavy chain α-myosin. 

The cardiac hypertrophic effects of FGF-23 are mediated by FGFR-dependent activation of the calcineurin nuclear factor of the activated T-cell (NFAT) signaling cascade, but do not require Klotho as a co-receptor [26]. 

As to Klotho, the exact pathophysiologic mechanisms behind its association with CKD progression are yet to be identified. This protein, which is mainly produced by the kidneys, consists of a long extracellular domain connected to a small intracellular C-terminal domain through a membrane-spanning segment. Its signaling properties depend upon the segment of the protein involved [7,28]. In fact, while membranous Klotho is a co-receptor for fibroblast growth factor 23 (FGF-23) along with the FGF receptor (FGFR), acting as a downregulator of phosphate metabolism, soluble Klotho results from the cleavage of the extracellular domain of the membranous primordial protein and exerts its action as a paracrine factor [7,8]. The specific actions of Klotho are still being investigated, but it appears that Klotho relies on its enzymatic properties to act as a major regulator of ion transport and growth factor signaling [8]. For instance, Klotho has been shown to reduce stress-induced cardiac hypertrophy by inhibiting cardiac TRPC6 channels in cardiomyocytes, as well as to stimulate TRPV5 receptors in the renal distal tubules, thus allowing the regulation of phosphate metabolism independently of calcium levels [9,10]. Its role in uremic myocardiopathy is not yet fully understood. Studies performed on animal models showed there is an association between ventricular hypertrophy and increased expression of TRPC6 (transient receptor potential canonical 6) channels, whose expression is regulated by a wide range of redundant mechanisms. Recently, Xie et al. demonstrated that Klotho may inhibit cardiac TRPC6 channels, thereby protecting the myocardium against excessive/pathological remodeling [7,29].

The elucidation of Klotho- and FGF-23-mediated functions in CKD paved the way for the development of targeted therapies. For instance, Di Marco et al. showed that the progression of cardiac hypertrophy and cardiac fibrosis could be reduced by antibody-mediated blockage of the FGF receptor [30], which would also help revert the CKD-associated mineral and bone disorders, as both the decrease in vitamin D and the increase in PTH are FGF-23-mediated [28,29,31]. On the other hand, Xie et al. observed that the administration of soluble Klotho to CKD patients prevented the progression of uremic cardiomyopathy [29] and Yin and collaborators proposed that kidney fibrosis might be abrogated by a fine-tuned control of the pro-fibrotic TGFβ functions [32]. At this point, it has become evident that a concerted plan of action that accommodates these new findings is urgently needed, aiming to provide CKD patients with a better diagnosis, follow up, prognosis and treatment.

In our study, the group with concentric hypertrophy had a higher cardiovascular mortality when compared to the group with eccentric hypertrophy and normal left ventricular geometry. Elevated levels of FGF-23 are implicated in elevated cardiovascular mortality in CKD when adjusted for the traditional risk factors and other mineral metabolism markers. The role of FGF-23 in cardiovascular mortality is not yet fully understood [33]. Pi et al. recently reported a relationship between the elevation of FGF-23 and activation of the renin-angiotensin-aldosterone system (RAAS). Activation of the RAAS through angiotensin II may explain this relationship between elevated FGF-23 and CV mortality [34].

Despite acknowledging the small sample size, the sparse number of cardiovascular events and the limited statistical power of these analyzes as the main limitations of this study, the current study presents preliminary results of a long term project. Notwithstanding this, the role that these new markers may play in cardiovascular morbidity and mortality in this particular population with diabetes and mild to moderate CKD is evident in the study.

Subsequent research is expected to evaluate the role of Klotho and FGF-23 in left ventricular diastolic dysfunction and progression to heart failure in patients with chronic renal disease.

## 4. Materials and Methods 

### 4.1. Subjects

This prospective analysis was conducted in the outpatient diabetic nephropathy (DN) clinic of the Centro Hospitalar Universitário do Algarve in Faro, Portugal, from 2012 to 2017, enrolling a total of 107 (female = 40) adult diabetic patients with stage 2–3 CKD. Before its implementation, the study was granted ethical approval by the Hospital’s Ethics Committee (November 2011), ensuring that all principles of the Declaration of Helsinki of 1973 (revised in 2000) were followed. All procedures were implemented after obtaining patients’ written informed consent. Diabetes was classified as per the American Diabetes Association guidelines [35].

The exclusion criteria were: age > 65 years; previous CVD–defined as a history of one or more of the following: non-fatal myocardial infarction, angina pectoris (stable or unstable), stroke or transient ischemic attacks, peripheral vascular disease or congestive heart failure; atrial fibrillation, severe coronary heart disease associated with the presence of a wall motion abnormality by two dimensional echocardiography, left bundle branch block, severe valvular heart disease confirmed with echocardiogram; changes in the GFR of more than 30% in the previous 3 months, changes in antihypertensive therapy 2 weeks before study, uncontrolled hypertension (BP ≥ 140/90 mmHg); albumin/creatinine ratio (ACR) ≥ 500 mg/g assessed on two occasions with an interval of 3 months; estimated glomerular filtration rate (eGFR) ≤ 29 mL/min/1.73 m^2^ or ≥ 89 mL/min/1.73 m^2^; parathyroid hormone (PTH) ≥ 350 pg/mL; phosphorus (Pi) > 5.5; type 1 diabetes; non-diabetic renal disease; neoplastic or infectious diseases.

Medical history, laboratory data, and current therapy were collected during the initial visit.

### 4.2. BP and Echocardiographic Measurements

The blood pressure (BP) was determined with oscillometric methods, with the patient in dorsal decubitus. Three measurements were taken with an interval of 5 min between them.

Echocardiographic evaluation was obtained using standard M mode and two-dimensional images (Vivid 7 Dimension-GE Healthcare Ultrasound; GE Healthcare, Waukesha, WI). Offline analysis was obtained using the workstation Echopac PC’08 version 7.0.0 GE Vingmed Ultrasound (GE Healthcare), and the measurements were consistently obtained by the same physician. Images were digitally stored and analyzed by two independent experienced cardiologists. Quality control procedures included blind rereading and patient reexamination to allow assessment of intra-reader variability, inter-reader variability, and intra-patient variability. An average of three measurements was made for each variable.

Patients were divided into three groups according to the left ventricular mass index (LVMI) and the relative wall thickness (RWT) values, calculated using the following equations:

(1) Regression equation from Penn convention:
LV mass = 1.04 ([LVIDD + PWTD + IVSTD]^3^ − [LVIDD]^3^) − 13.6 g(1)
where LVIDD = left ventricular internal diameter in diastole, PWTD = posterior wall thickness in diastole and IVSRD = inter-ventricular septum thickness in diastole. LVMI was then obtained by dividing LV mass by body surface area.
RWT = 2(PWT)/LVDD(2)
where PWT is the posterior wall thickness and LVDD is the left ventricle diastolic diameter. 

LVH and RWT were used to categorize LV geometry: normal (no LVH and normal RWT), eccentric hypertrophy (LVH and normal RWT), and concentric hypertrophy (LVH and increased RWT). Patients were classified as having LVH if they had a left ventricular mass index (LVMI)of 100 g/m^2^ in women and 131 g/m^2^ in men [36], and RWT was considered to be increased if it was ≥0.45 according to the study by Chen et al. [37].

### 4.3. Follow-up

The Centro Hospitalar Universitário do Algarve – Unidade de Faro outpatients’ DN clinic is the Algarve’s reference center for nephrology and guarantees a regular follow-up for patients with DN. Patients with a more severe condition are monitored every 3 months and those with a milder disease every 6 months. Mean duration of follow-up was 33.8 ± 8.6 months. 

### 4.4. Blood Measurements 

Fasting samples were drawn from all subjects and plasma was frozen at −80 °C in order to measure eGFR, phosphorus (Pi), calcium (Ca), parathyroid hormone (PTH), glycated hemoglobin (HbA1c), insulin resistance index, interleukin-6 (IL-6), fibroblast growth factor-23 (FGF23), 1,25(OH)_2_D3 (vitamin D), soluble α-Klotho (Klotho) and oxidized low density lipoprotein (oxLDL). Serum levels of vitamin D (1.25 dihydroxicolecalciferol) were determined by radioimmunoassay (IDS, Boldon, UK). The vitamin D levels were the average of three measurements conducted in winter, spring and summer. Serum levels of FGF23 were quantified using the enzyme-linked immunosorbent assay *Human FGF-23 (C-Term)* enzyme-linked immunoassay (ELISA) kit (Cat. #60-6100 ImmunotopicsInc, San Clemente, CA, USA). Soluble α-Klotho serum levels were determined by ELISA using the Human soluble α-Klotho kit (Code no- 27998, IBL – Immuno-Biologial Laboratories Co., Ltd., Gunma, Japan), by adapting the manufacturer’s instructions to the Triturus microplate automatic system (Grifols S.A., Barcelona, Spain). Two blood samples were collected for the validation of the circadian variability of Klotho and FGF-23, P and Ca were assayed by the ARCHITECT c and the AEROSET Systems (Abbott Diagnostics Division, Abbott Laboratories Abbott Park, IL). IL-6 and oxLDL determination was performed with a sandwich ELISA kit (eBioscience, San Diego, CA, USA). HbA1c and PTH levels were measured using a spectrophotometry technique and electrochemiluminescent immunoassay (ECLIA), respectively. PTH concentrations were measured on an Immulite 2000 Intact PTH assay (Cat. #L2KPP2, Siemens Medical Solutions Diagnostics, Los Angeles, CA, USA). The degree of insulin resistance was estimated using the homeostasis model assessment (HOMA-IR) described by Matthews et al. [38]. Serum creatinine was assayed by the multigent creatinine enzymatic method, using the ARCHITECT^®^ device (Abbott Diagnostics Division, Abbott Laboratories Abbott Park, IL; USA). We estimated the GFR according to the Chronic Kidney Disease Epidemiology Collaboration (CKD-EPI) Equation [39].

### 5.5. Urine Measurements

The ACR was determined from a first-morning urine collected on the 1st and the 8th day of the inclusion period. Urinary albumin and creatinine were determined immediately using the DCA 2000^®^ albumin/creatinine assay system (Bayer Diagnostica, Barcelona, Spain) kit. A laboratory immunoturbidimetric assay was used for albumin and creatinine determinations and the results are reported as ACR (μg/mg). Intra individual variation of ACR was assessed in 8 of the 16 subjects for 7 consecutive days.

### 4.6. Cardiovascular Events 

The primary cardiovascular outcome of this study was cardiovascular mortality. Deaths were confirmed by review of autopsy reports, death certificates, medical records, or information obtained from the next of kin or family members and classified, according to its cause, as cardiovascular or non-cardiovascular death. Cardiovascular deaths were defined as caused by coronary heart disease, heart failure, peripheral vascular disease and cerebrovascular disease.

The secondary outcome evaluated the predictive factors of cardiovascular hospitalizations/admissions. These were classified upon discharge, considering only admissions caused by coronary heart disease (myocardial infarction, stable or unstable angina pectoris), congestive heart failure, peripheral vascular disease and cerebrovascular disease (stroke or transient ischemic attacks) based on recent international guidelines (Kavousi, Leening, Nanchen, et al., 2014) [40]. According to the presence or absence of cardiovascular hospital admission during the study period, our population was divided into two groups: G-1—with cardiovascular hospitalization (*n* = 26), and G-2—without cardiovascular hospitalization (*n* = 81).

### 4.7. Statistical Analysis

Continuous variables are presented as mean and standard deviation and categorical variables are expressed as percentages and compared using the chi-squared test. Demographic and clinical factors associated with LVH were evaluated by ANOVA and the Kruskal–Wallis H test, respectively. Patients’ demographic and clinical characteristics at baseline were grouped according to their level of cardiac hypertrophy.

Simple linear regressions were used to investigate possible correlations between these variables and LVMI, while multivariable nominal logistic regression identified factors associated to eccentric and concentric hypertrophy, using as a reference, patients with normal geometry.

We evaluated time to fatal CV end point by Kaplan–Meier survival analysis compared by log-rank test among the three patterns of LV geometry for measuring patients’ survival rate and a comparison between the three groups was based on the log rank test. The null hypothesis was rejected below the 5% level. Statistical analysis was performed using SPSS (version 17.0; SPSS, Chicago, IL, USA).

The generalized linear model (GLM) for binary dependent variables was used (binomial distribution) with a logistic link function. The exponentials of the model parameters were the adjusted odds ratio (ORa) to other variables of the model. The test was used to determine whether Klotho, FGF-23 and cardiac geometry were important predictive factors in the determination of cardiovascular hospitalization. Prior to conducting the GLM, we calculated the median level of Klotho serum and FGF-23 in the current study population. The subjects were categorized in the following classes: Klotho (≥313 pg/mL or <313 pg/mL), and FGF-23 (≥168 RU/mL or <168 RU/mL). 

The independent variables associated with cardiovascular mortality were identified by Cox regression models. Variables included in these models were: age (<65; ≥65), gender, median: PTH (<138; ≥138), Pi (<3.6; ≥3.6), 1.25(OH)2D3 (≥21;< 21), α-klotho (<313; ≥313), FGF-23 (<168; ≥168), ACR (<181; ≥181), IL-6 (<6; ≥6), stratified renal disease (60–89-stage 2; 30–59 stage 3) and cardiac geometric classification (normal; eccentric hypertrophy; concentric hypertrophy).

We used the receiver operating characteristic (ROC) curve to identify klotho and FGF-23 cut-offs associated with the highest cardiovascular death in this population.

## 5. Conclusions

The role of emerging factors like FGF-23 and Klotho in cardiovascular risk in both the early and late stages of chronic kidney disease is not entirely understood. The entire process involves direct and indirect mechanisms that contribute to this high cardiovascular risk. Moreover, these proteins seem to have independent roles in the pathophysiology of cardiovascular disease and CKD and have recently been considered relevant potential therapeutic agents in this disease. 

The early identification of CKD onset and risk stratification of CKD progression and/or CKD-related complications are essential for early treatment of CKD patients to ameliorate their comorbidity burden, particularly CVD, and prevent the development of ESRD.

## Figures and Tables

**Figure 1 ijms-20-01536-f001:**
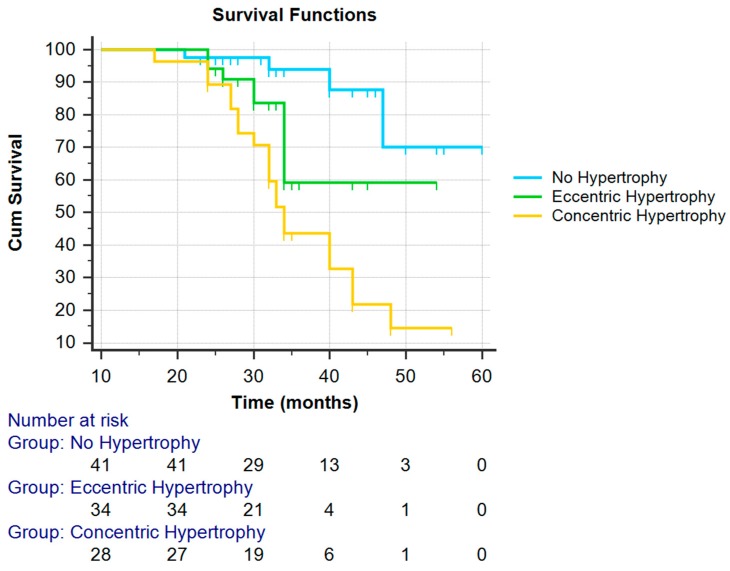
Cardiovascular mortality according to left ventricular geometry (Kaplan–Meier Analysis).

**Table 1 ijms-20-01536-t001:** Patient demographic and clinical characteristics at baseline.

Characteristics	Values
Number of patients, *n*	107
Gender (f/m)	40/67
BMI (Kg/m^2^)	26.04
Age (years)	57.19 ± 7.05
Hb (g/dL)	12.97 ± 1.83
Albumin (g/dL)	4.27 ± 0.48
eGFR (mL/min)	52.89 ± 20.15
ACR (µg/mg)	181.89 ± 33.83
Pi (mg/dL)	3.99 ± 0.85
PTH (pg/mL)	113.11 ± 74.65
Calcium (mg/dL)	9.48 ± 0.68
FGF-23 (RU/mL)	135.04 ± 135.23
1,25 (OH)2 Vitamin D (pg/mL)	21.21 ± 7.36
α-Klotho (pg/mL)	331.10 ± 171.06
IL-6 (pg/mL)	5.71 ± 3.80
Oxidized LDL (U/L)	39.91 ± 19.55
HOMA-IR	1.84 ± 1.67
HbA1c (%)	7.67 ± 1.47
LVMI (g/m^2^)	99.31 ± 23.45
Systolic BP (mmHg)	126.59 ± 16.53
Diastolic BP (mmHg)	78.58 ± 9.98
Diabetes-related CKD evolution time (months)	73.8 ± 8.7

Values are presented as mean ± standard deviation.

**Table 2 ijms-20-01536-t002:** Patient demographic and clinical characteristics at baseline according to level of cardiac hypertrophy.

	Group 1 Normal Geometry *n* = 41	Group 2 Eccentric Hypertrophy *n* = 38	Group 3 Concentric Hypertrophy *n* = 28	*p*
Age (years)	56.12 ± 7.35	57.29 ± 7.27	58.61 ± 6.21	NS
eGFR (mL/min/1.73 m^2^)	62.89 ± 20.58	47.13 ± 15.18	46.06 ± 20.10	<0.001
BMI (Kg/m^2^)	26.3 ± 1.08	26.8 ± 0.42	27.2 ± 0.64	NS
Hb (g/dL)	13.57 ± 1.73	12.98 ± 1.81	12.98 ± 1.81	0.003
Pi (mg/dL)	3.32 ± 0.64	4.76 ± 0.54	4.13 ± 0.75	<0.001
PTH (pg/mL)	74.16 ± 49.67	104.07 ± 63.26	182.41 ± 73.07	<0.001
ACR (µg/mg)	122.13 ± 90.60	193.50 ± 144.86	253.64 ± 91.30	<0.001
IL-6 (pg/mL)	2.93 ± 1.82	8.47 ± 3.22	10.09 ± 2.42	<0.001
α-Klotho (pg/mL)	399.17 ± 143.72	344.82 ± 53.52	160.82 ± 60.70	<0.001
FGF-23 (RU/mL)	66.75 ± 39.66	113.06 ± 54.20	264.84 ± 72.96	<0.001
[1,25(OH)_2_D3] (pg/mL)	25.78 ± 4.49	21.88 ± 6.80	18.60 ± 5.22	<0.001
Systolic BP (mmHg)	124.2 ± 14.2	121.1 ± 15	125.5 ± 16.5	0.057
Diastolic BP (mmHg)	71.2 ± 10.4	71.9 ± 11.8	69.6 ± 11.6	NS
Heart rate (bpm)	67 ± 12	70 ± 13	69 ± 11	NS
LV end-diastolic volume (mL)	120 ± 17	162 ± 32	118 ± 20	0.005
LV end-systolic volume (mL)	42 ± 10	62 ± 23^x^	45 ± 15	<0.001
LV mass/body surface area (g/m^2^)	96 ± 11	132 ± 22	134 ± 20	NS
RWT	0.38 ± 0.3	0.37 ± 0.06	0.46 ± 0.03	NS
RAS inhibitor/or ACEI (%)	78.7	80.7	88.9	NS
Calcium channel blockers with renoprotective action (%)	35.6	48.6	52.3	NS
Diabetes-related CKD evolution time (months)	75.5 ± 9.6	71.7 ± 7.3	74.1 ± 8.7	NS

LV, left ventricular; LVH, left ventricular hypertrophy; RWT, relative wall thickness; ARB, angiotensin-converting inhibitor; ACEI, angiotensin-converting enzyme inhibitor; NS means without statistical significance.

**Table 3 ijms-20-01536-t003:** Simple linear regression analysis between LVMI and other parameters.

Variable	*R*	*p*-Value
Gender	0.741	0.354
Age	0.243	0.223
BMI	0.480	0.650
eGFR	−0.227	0.060
Hb	−0.317	0.973
Pi	0.672	0.005
PTH	0.520	0.199
ACR	0.427	0.075
IL-6	0.737	0.087
α-Klotho	−0.440	0.0001
FGF-23	0.622	0.0001

**Table 4 ijms-20-01536-t004:** Clinical characteristics’ correlation with eccentric or concentric left ventricular hypertrophy by multinomial logistic regression analysis.

Variable	Eccentric Hypertrophy			Concentric Hypertrophy		
	OR	95% CI	*p*	OR	95% CI	*p*
α-Klotho	1.005	0.97–1013	0.186	0.737	0.603–0.972	0.031
Gender	1.670	0.341–8.174	0.527	1.986	0.905–2.009	0.063
Age	1.095	0.970–1.235	0.142	1.053	0.821–1.349	0.685
eGFR	0.959	0.921–0.999	0.043	0.769	0.622–1.050	0.085
Pi	2.859	2.238–5.693	0.003	1.115	0.958–2.318	0.063
PTH	0.980	0.960–1.001	0.058	0.994	0.969–1.020	0.664
ACR	1.005	0.999–1.011	0.131	0.977	0.945–1.011	0.184
FGF-23	1.008	0.996–1.020	0.219	1.031	1.008–1.205	0.009
[1,25 (OH)2 D3]	0.904	0.726–1.126	0.366	0.529	0.263–1.061	0.073
Hb	1.040	0.686–1.577	0.852	0.307	0.295–1.001	0.068
IL-6	1.255	0.836–1.885	0.274	1.443	0.984–2.858	0.208

*Reference category: No Hypertrophy.

**Table 5 ijms-20-01536-t005:** Predictive factors of cardiovascular hospitalization.

	Initial Model			Optimized Model		
	OR_a_	95%CI for OR_a_	*p*	OR_a_	95%CI for OR_a_	*p*
α-Klotho groups						
<313	1.491	1.207–2.125	0.014	11.320	1.061–1.456	0.024
≥313	Ref.			Ref.		
FGF-23 groups						
≥168	1.689	1.004–3.500	0.004	1.105	1.000–1.763	0.012
<168	Ref.			Ref.		
Concentric Hypertrophy	4.889	3.372–5.512	0.023	2.284	1.970–4.720	0.1
Eccentric Hypertrophy	3.360	0.692–4.890	0.064	1.410	0.984–2.523	0.075
No Hypertrophy	Ref.			Ref.		
*p*-value (model)	0.007			<0.001		
Area under ROC (*p*-value)	-			0.980 (*p* < 0.001)		

ORa: Adjusted odds ratio, 95%CI for OR: 95% confidence interval for the odds ratio, Ref: Category versus the one is making comparisons.

**Table 6 ijms-20-01536-t006:** Factors independently associated with cardiovascular mortality.

	HR_a_	95%CI for OR_a_	*p*-Value
Pi			
<3.6	Ref.		
≥3.6	1.079	1.015–3.409	0.025
α-Klotho groups			
<313	2.377	1.488–11.585	0.044
≥313	Ref.		
FGF-23 groups			
≥168	2.046	1.008–8.249	0.014
< 168	Ref.		
Concentric	3.254	1.035–6.699	0.041
Eccentric	1.112	1.070–3.850	0.050
No Hypertrophy	Ref.		

HRa: Adjusted hazards ratio, 95%CI for HR: 95% confidence interval for the hazard ratio, Ref: Category that is being used to make comparisons against. The other variables included in the model, but that presented no statistical significance were: age, gender, PTH, 1.25(OH)2D3, Hg, ACR, IL-6, stratified renal disease stage 2 (60–89) and 3 (30–59).

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
