# Peer review of "Plasmatic Klotho and FGF23 Levels as Biomarkers of CKD-Associated Cardiac Disease in Type 2 Diabetic Patients"

_ijms, 2019, doi:10.3390/ijms20071536_

Reviewer 1 Report

General comments

In this study, authors presented a cohort of diabetic patients (mostly male and type 2) with 2-3 stage CKD and without previous cardiovascular disease (CVD) and they divided them into three groups according to LVMI and RWT. Patients were followed by the mean duration of 33.8 months. Main results were that low klotho and high fibroblast growth factor-23 (FGF-23) were associated with a greater risk of concentric hypertrophy in multinomial regression while klotho, FGF-23 and cardiac geometry were independently associated with CVD-related hospitalization.  Fatal CV events were significantly associated with eccentric hypertrophy, concentric hypertrophy, elevated serum phosphate levels, elevated FGF-23 and decreased alpha klotho. Finally, the authors concluded that in this diabetic cohort, Klotho and FGF-23 were associated with cardiovascular risks in the early stages of CKD (stages 2-3).

Ethical/legal disclosures

A study was approved by the local Hospital's Ethics Committee and legal disclosures are described appropriately.

Statistical methodology

xxx

Specific comments

Title: I would suggest authors change the title and discard expressions such as „valuable“ etc. because such terms are pseudoscientific and do not convey an important message. Authors should also add that Klotho and FGF23 levels were determined in CKD-associated cardiac disease IN diabetic patients or diabetic type 2 patients. This should be stated clearly in the title since the current title does not provide an adequate and precise description of the presented population

CKD stages 2 to 3 by definition is mild to moderate renal disease. Authors should elaborate on why they did not include diabetic patients with more advanced stages of kidney disease? In many contemporary studies, significant renal disease is defined as at least CKD 3 and, of course, higher classes being defined as severe and end-stage renal disease

Line 76 – fix GFR 30% - I guess authors wanted to state „changes in the GFR of more than 30% in the previous 3 months“

Line 88 – change „imagens“ to „images“

What was the median follow-up period?

Please disregard term „risk“ when discussing results obtained from regression analysis in which you report „odds ratio“. Those are odds or likelihood of an event. A term risk should be used only if your results were reported in absolute or relative risk, which clearly is not a case here

What about anthropometric data such as waist-to-hip ratio (WHR) or body-mass index (BMI)? Were these collected among your patients? If yes, these should be included in the simple regression analysis when analyzing correlations and also in terms of determining differences between groups of interest. In my experience, BMI, WHR and central adiposity (visceral fat) can play an important role in the metabolism and biotransformation of these biomarkers, and this is especially the case in smaller sample sizes as presented in this study. BMI and additional anthropometric parameters beyond age and sex only can significantly skew results and all analyses should be at least age, sex, and BMI/WHR-adjusted. Since LVMI calculation requires some of these anthropometric measures, they should be included/discussed in the manuscript

Lines 236-239. Authors should state explicitly how many death events occurred in each group by LV geometry, not just the percentage

Authors state 90.2% survival in patients without hypertrophy at 60 months, followed by 76.3% in eccentric hypertrophy and only 53.6% in patients with concentric hypertrophy. This is quite high mortality for concentric hypertrophy group, especially given that this is population without previous CVD event, mild-to-moderate CKD and many other exclusions. Authors should discuss the main causes of death in these populations and elaborate on these results

Lines 244-256 – Authors now use odds ratio instead of risks. Please correct these inconsistencies throughout the manuscript

Which parameters/covariates were included in the Cox regression model (Table 6)

Very importantly, what was the LV ejection fraction among compared groups? LVEF should be included in all models and comparisons, therefore, I would strongly advise authors to include it in future revision and analyses. I think that any discussion about LV parameters without taking LVEF into account is not a good approach and might significantly skew the interpretation of the data. It would be also interesting to see if FGF-23 levels correlate with systolic function (e.g. LVEF)

Authors have not discussed or elaborated on any potential weaknesses of this study. For example, a low number of cases, a low number of patients represented in subgroup analyses might have limited the interpretation of the results, etc. Authors should definitely comment on weaknesses and employ a proactive approach

Author Response

General comments

In this study, authors presented a cohort of diabetic patients (mostly male and type 2) with 2-3 stage CKD and without previous cardiovascular disease (CVD) and they divided them into three groups according to LVMI and RWT. Patients were followed by the mean duration of 33.8 months. Main results were that low klotho and high fibroblast growth factor-23 (FGF-23) were associated with a greater risk of concentric hypertrophy in multinomial regression while klotho, FGF-23 and cardiac geometry were independently associated with CVD-related hospitalization.  Fatal CV events were significantly associated with eccentric hypertrophy, concentric hypertrophy, elevated serum phosphate levels, elevated FGF-23 and decreased alpha klotho. Finally, the authors concluded that in this diabetic cohort, Klotho and FGF-23 were associated with cardiovascular risks in the early stages of CKD (stages 2-3).

Specific comments

Title: I would suggest authors change the title and discard expressions such as „valuable“ etc. because such terms are pseudoscientific and do not convey an important message. Authors should also add that Klotho and FGF23 levels were determined in CKD-associated cardiac disease IN diabetic patients or diabetic type 2 patients. This should be stated clearly in the title since the current title does not provide an adequate and precise description of the presented population

Response: DONE

CKD stages 2 to 3 by definition is mild to moderate renal disease. Authors should elaborate on why they did not include diabetic patients with more advanced stages of kidney disease? In many contemporary studies, significant renal disease is defined as at least CKD 3 and, of course, higher classes being defined as severe and end-stage renal disease

Response: Wahl P et al, : Earlier Onset and Greater Severity of Disordered Mineral Metabolism in Diabetic Patients With Chronic Kidney Disease. Diabetes Care 2012 May; 35(5): 994-1001  » demonstrated that in the early stages of renal disease in diabetics patients higher FGF-23 levels were found when compared to non-diabetic CKD patients . Therefore, we sought to ascertain whether higher FGF-23 levels in the early stages (stage 2 and 3) will influence the myocardial structure and, consequently, cardiovascular morbidity and mortality. Moreover, some studies have already demonstrated that in later stages there is an increased cardiovascular events in patients with lower klotho and/or increased FGF-23 levels.

Line 76 – fix GFR 30% - I guess authors wanted to state „changes in the GFR of more than 30% in the previous 3 months“

Response: DONE

Line 88 – change „imagens“ to „images“

Response: DONE

What was the median follow-up period?

Response: The median is 33 months.

Please disregard term „risk“ when discussing results obtained from regression analysis in which you report „odds ratio“. Those are odds or likelihood of an event. A term risk should be used only if your results were reported in absolute or relative risk, which clearly is not a case here

Response: DONE

What about anthropometric data such as waist-to-hip ratio (WHR) or body-mass index (BMI)? Were these collected among your patients? If yes, these should be included in the simple regression analysis when analyzing correlations and also in terms of determining differences between groups of interest. In my experience, BMI, WHR and central adiposity (visceral fat) can play an important role in the metabolism and biotransformation of these biomarkers, and this is especially the case in smaller sample sizes as presented in this study. BMI and additional anthropometric parameters beyond age and sex only can significantly skew results and all analyses should be at least age, sex, and BMI/WHR-adjusted. Since LVMI calculation requires some of these anthropometric measures, they should be included/discussed in the manuscript

Response: DONE

Lines 236-239. Authors should state explicitly how many death events occurred in each group by LV geometry, not just the percentage

Response: DONE

Authors state 90.2% survival in patients without hypertrophy at 60 months, followed by 76.3% in eccentric hypertrophy and only 53.6% in patients with concentric hypertrophy. This is quite high mortality for concentric hypertrophy group, especially given that this is population without previous CVD event, mild-to-moderate CKD and many other exclusions. Authors should discuss the main causes of death in these populations and elaborate on these results

Response: DONE

Lines 244-256 – Authors now use odds ratio instead of risks. Please correct these inconsistencies throughout the manuscript

Response: DONE

Which parameters/covariates were included in the Cox regression model (Table 6)

Response: The variables included in Cox model were:gender; age; eGFR; Pi, PTH; ACR; [1,25 (OH)2 D3], Hb; IL-6; Klotho, FGF-23; Concentric Hypertrophy; eccentric hypertrophy; no hypertrophy and renal disease stage

Very importantly, what was the LV ejection fraction among compared groups? LVEF should be included in all models and comparisons, therefore, I would strongly advise authors to include it in future revision and analyses. I think that any discussion about LV parameters without taking LVEF into account is not a good approach and might significantly skew the interpretation of the data. It would be also interesting to see if FGF-23 levels correlate with systolic function (e.g. LVEF)

Response: I appreciate your suggestion.

In our study, the ejection fraction was not evaluated because its relation with the markers we are evaluating   is not consensual:  Almahmoud et al. Fibroblast Growth Factor23 and Heart Failure With Reduced Versus Preserved Ejection Fraction: MESA. J Am Heart Assoc. 2018 Sep 18; 7(18): In this study, we found no significant association between higher FGF23 and HFrEF events, while previous studies suggested an association of higher FGF23 concentrations with lower LVEF. However, these studies showed very modest association with reduced EF. For example, in a study by Agarwal et al, the difference in mean EF between the lowest and highest quartiles of FGF23 was negligible (62.2% versus 59.7%, respectively) and was still considered as normal EF. Furthermore, in a postkidney transplant cohort, no significant association was found between FGF23 and LVEF

However in a future work we intend to evaluate the relationship of FGF23;klotho; resistin, visfatin, apelin; not only left ventricular geometric changes, LVEF, cardiac insufficiency criteria according to the new criteria for CI 2016, and changes in diastolic function.

Authors have not discussed or elaborated on any potential weaknesses of this study. For example, a low number of cases, a low number of patients represented in subgroup analyses might have limited the interpretation of the results, etc. Authors should definitely comment on weaknesses and employ a proactive approach

Response: DONE

Reviewer 2 Report

I have just some minor comments

Report the number of excluded patients with the reasons

Sudden death has to be included as CV death

Add patients at risk for each group in Kaplan-Meier curve

Author Response

 Report the number of excluded patients with the reasons

Response: The number of patients excluded from our sample were 175 type 2 diabetics, for the following causes:

- 75 refused to participate in the study

- 50 had at baseline a GFR between 20-15 ml /

- 25 presented significant alterations in mineral metabolism were under treatment for hyperparathyroidism and hyperphosphatemia.

- 25 of our patients had heart failure criteria according to the IC 2016 criteria.

In our study, we intend to evaluate the role of FGF23 and Klotho in the early stages of chronic kidney disease as well as in cardiovascular alterations aiming to identify risk factors to be modified as well as adequate therapeutic and educational intervention to contribute to a better quality of life

Sudden death has to be included as CV death

Response: DONE

Add patients at risk for each group in Kaplan-Meier curve

Response: DONE

Reviewer 3 Report

Silva and colleagues present a prospective study about the potential use of plasmatic Klotho and FGF-23 as biomarkers for Chronic Kidney Disease (CKD)-associated cardiac disease and mortality. In particular, they conclude that CKD patients with low serum Klotho levels (313 pg/mL) and high serum FGF-23 levels (168 pg/mL) are at high risk of adverse cardiovascular outcomes.

The paper is well-written and very interesting. However, the authors might consider the following comments:

At line 84 the title of the paragraph is “BP and Echocardiographic Measurements”, but they do not mention at all the method used to measure the blood pressure of patients: auscultatory or by means oscillometric devices?

The authors report the age of the patients in the Tables but never their weight. It is important to know if some patients are obese. In fact, many studies suggest that indexing the LV mass to height raised to allometric power of 2.7 has advantages over indexing to BSA (as authors have done), especially when attempting to predict events in obese patients.

At lines 106 and 107 they wrote: Patients were classified as having LVH if they had LVMI.100 g/m2in women and .131 g/m2in men, and RWT was considered to be increased if 0.45 according to the study by Chen at al. It is not clear LVMI.100 g/m2 and .131 g/m2which cut-off values are. What does that . before the number mean? Did they want to put a zero before the . ? But in this case the values are wrong. Moreover, if the cut-off values are 100 g/m2in women and 131 g/m2in men, are not the same cited in the articles that they consider as references.

Among the exclusion criteria from the study, the authors report “severe valvular heart disease as assessed by clinical examination” (Lines 75-76). I think it is quite difficult to make a precise diagnosis of valvulopathy based only on the clinical examination and not by an echocardiography.

The authors should review carefully the text because there are some mistakes such as:

Line 88 - Imagens: images

Line 120 - 1,25-dihroxicolecalciferol: 1,25-dihydroxicolecalciferol

Line 307 - (168 pg/mL): (168 pg/mL)

Author Response

At line 84 the title of the paragraph is “BP and Echocardiographic Measurements”, but they do not mention at all the method used to measure the blood pressure of patients: auscultatory or by means oscillometric devices?

Response: DONE

The authors report the age of the patients in the Tables but never their weight. It is important to know if some patients are obese. In fact, many studies suggest that indexing the LV mass to height raised to allometric power of 2.7 has advantages over indexing to BSA (as authors have done), especially when attempting to predict events in obese patients.

Response: Thanks for the suggestion, however we opted for the Index of body mass because of the work in this area consider a good anthropometric indicator.

DONE

At lines 106 and 107 they wrote: Patients were classified as having LVH if they had LVMI.100 g/m2in women and .131 g/m2in men, and RWT was considered to be increased if 0.45 according to the study by Chen at al. It is not clear LVMI.100 g/m2 and .131 g/m2which cut-off values are. What does that . before the number mean? Did they want to put a zero before the . ? But in this case the values are wrong. Moreover, if the cut-off values are 100 g/m2in women and 131 g/m2in men, are not the same cited in the articles that they consider as references.

Response: Corrected reference :Savage DD, Garrison RY, Kannel WB, Levy D, et al. The spectrum of left ventricular hypertrophy in a general population sample: the Framingham study. Circulation. 1987; 75 (suppl I):26-33.

In this study the Echo LVH was considered to be present if the left ventricular mass index exceeded 131 g/m2for men and 100 g/m2women. In our study we used the same cut-off.

 Did they want to put a zero before the--?

Response: Done

not the same cited in the articles that they consider as references. DONE

Among the exclusion criteria from the study, the authors report “severe valvular heart disease as assessed by clinical examination” (Lines 75-76). I think it is quite difficult to make a precise diagnosis of valvulopathy based only on the clinical examination and not by an echocardiography.

Response: DONE

The authors should review carefully the text because there are some mistakes such as:

Line 88 - Imagens: images

DONE

Line 120 - 1,25-dihroxicolecalciferol: 1,25-dihydroxicolecalciferol

Response: DONE

Line 307 - (168 pg/mL): (168 pg/mL)

Response: DONE

Round  2

Reviewer 1 Report

Authors have answered my concerns effectively.